# High-throughput sequencing defines donor and recipient HLA B-cell epitope frequencies for prospective matching in transplantation

Jenny N. Tran [1], Oliver P. Günther [2], Karen R. Sherwood[1], Franz Fenninger[1], Lenka L. Allan[1], James Lan[1,3], Ruth Sapir-Pichhadze[4], Rene Duquesnoy[5], Frans Claas[6], Steven G. E. Marsh [7], W. Robert McMaster[8,9], Paul A. Keown [1,3,9 ✉] & Genome Canada Transplant Consortium*

Compatibility for human leukocyte antigen (HLA) genes between transplant donors and recipients improves graft survival but prospective matching is rarely performed due to the vast heterogeneity of this gene complex. To reduce complexity, we have combined next-generation sequencing and in silico mapping to determine transplant population frequencies and matching probabilities of 150 antibody-binding eplets across all 11 classical HLA genes in 2000 ethnically heterogeneous renal patients and donors. We show that eplets are more common and uniformly distributed between donors and recipients than the respective HLA isoforms. Simulations of targeted eplet matching shows that a high degree of overall compatibility, and perfect identity at the clinically important HLA class II loci, can be obtained within a patient waiting list of approximately 250 subjects. Internal epitope-based allocation is thus feasible for most major renal transplant programs, while regional or national sharing may be required for other solid organs.

[1] Department of Pathology and Laboratory Medicine, University of British Columbia, Vancouver, BC, Canada. [2] Günther Analytics, Vancouver, BC, Canada. [3] Department of Medicine, University of British Columbia, Vancouver, BC, Canada. [4] Department of Medicine, McGill University and MU-HRI, Montreal, QC, Canada. [5] Department of Pathology, University of Pittsburgh, Pennsylvania, PA, USA. [6] Department of Immunohematology and Blood Transfusion, University of Leiden, Leiden, Netherlands. [7] Anthony Nolan Research Institute and UCL Cancer Institute, Royal Free Campus, London, UK. [8] Department of Medical Genetics, University of British Columbia, Vancouver, BC, Canada. [9] Infection and Immunity Research Centre, University of British Columbia, Vancouver, BC, Canada. *A list of authors and their affiliations appears at the end of the paper. ✉email: paul.keown@ubc.ca

Transplantation is the treatment of choice for irreversible renal failure, offering superior survival, quality of life, and economic costs compared to alternative options[1,2]. But despite superb initial success (1-year kidney graft survival often exceeds 95%), many grafts fail within the first decade[3]. While several factors may jeopardize the transplanted organ, graft rejection remains the overwhelming cause of failure[4,5] and antibody-mediated rejection (AMR) is the most serious and destructive form of this injury. It may occur early or late in the transplant course, presenting a spectrum that ranges from common acute and fulminant graft injury to chronic and progressive graft destruction. There is currently no effective therapy to reverse AMR, so measures to prevent this complication by reducing immunogenicity and modulating immunity are vital.

Human leukocyte antigen (HLA) genes are the most polymorphic in the human genome with over 25,000 alleles now identified[6]. These genes code for highly immunogenic HLA protein isoforms expressed on nucleated cells and are known to be the principal transplantation antigens and primary targets of graft rejection[7]. Compatibility for HLA genes between donor organs and graft recipients ensures excellent outcome in live donor transplantation, and improves graft survival in deceased donor transplantation[8,9], but is difficult to achieve due to the heterogeneity of this gene complex.

Discrete motifs on these proteins are central to both antigen recognition and response. Structural epitopes, clusters of amino acids on the surface of the HLA protein isoforms that are accessible to and are bound by antibody, encompass smaller eplets lying in a 3 Angstrom radius containing at least one polymorphic amino acid, that interact directly with the antibody paratope[10]. The quantitative mismatch of these eplets between donor and recipient provides an index of the risk of rejection, though not all epitope mismatches may be of equal importance and those occurring at certain HLA class II gene loci may be particularly critical. For example, Wiebe et al. have shown that HLA-DRB1 and -DQB1 mismatches are independent predictors of de novo class II donor-specific antibody development[11], and a nested case–control study found that the odds ratio for transplant glomerulopathy increases incrementally with increasing HLA-DR and -DQ eplet mismatches[12].

Quantitative epitope mismatch analysis is normally proposed as a post-hoc measure to predict rejection risk and to guide immunosuppressive treatment. However, prospective use of eplet matching to guide recipient selection offers a novel method to actively reduce donor immunogenicity, and the limited number of eplets may enable efficient matching. To determine whether this prospective strategy is clinically feasible for patients awaiting transplantation requires precise population data on donor and recipient frequency distributions. Here we present the first large-scale data comparing human HLA allele and eplet frequencies defined by high-resolution next-generation sequencing (NGS) in a heterogeneous transplant population, focusing on antibody-verified HLA eplets in light of their proven target role. We describe combinatorial epitypes comprising the array of eplets expressed by the HLA isoforms on each individual donor and recipient, and model the probabilities of achieving eplet identity at all or individual HLA gene loci to confirm the feasibility of prospective eplet matching within a national transplant program.

## Results

**Transplant patients and donors.** A total of 2000 subjects from the BC renal transplant program had NGS sequence data at all 11 allelic HLA loci for the study. Of these, 154 subjects expressed alleles that were not yet present in the HLAMatchmaker database (47 alleles, average carrier rate ≤0.18%, Supplementary Table 1)

and were excluded from this analysis. Carrier rates of other HLA alleles in these subjects were otherwise comparable to the overall study population (Supplementary Fig. 1). The remaining 1846 subjects included 1049 patients with kidney failure and 797 kidney donors. Patient and donor groups were 62 and 52% males, with a mean age of 56 and 48 years, respectively. Four subgroups were included to control for bias, comprising patients prior to ($n = 611$) or post-transplant ($n = 438$), and deceased ($n = 243$) or living donors ($n = 554$).

**Visualization of eplets on HLA proteins.** Eplets were mapped onto 3D HLA protein structures for class I and class II (Fig. 1)[13–15]. In Fig. 1a, b HLA-A*02:01 is depicted from top-down and side views, respectively. Eplets existed in all extracellular regions, predominantly in the alpha helices around the peptide-binding groove encoded by the key exons 2 and 3, but also outside this region in alpha 3 encoded by non-key exon 4. Figure 1c, d represents HLA-DQ2.3 protein, encoded by HLA-DQA*03:01 and HLA-DQB1*02:01. Eplets were found in the alpha helices of both alpha and beta chains, and within the beta sheet of the beta chain. Visualization of the side of the protein show eplets in beta 2 (encoded by non-key exon 3) of the beta chain. Thus, the majority of eplets are found in the peptide-binding region but eplets exist outside this area, showing that antibody binding can occur across the entire extracellular portion of the HLA protein.

**Converting HLA alleles into eplets.** The 564 class I and 290 class II alleles in HLAMatchmaker were used to define a string of eplets for each allele. The network diagrams in Fig. 2 show the extensive sharing of eplets by alleles within and between HLA class I A, B, and C gene loci, while class II eplets were shared only by alleles within the same gene, except for DRB1, DRB3, DRB4, and DRB5. Interestingly, DPA1 contained mutually exclusive allele groups of eplet expression. For example, DPA1 alleles exclusively expressed either 50QA (encoded by DPA1*01:03, DPA1*01:04, DPA1*01:05, DPA1*01:06, DPA1*03:01, DPA1*03:02, and DPA1*03:03) or 50RA (encoded by DPA1*01:08, DPA1*02:01, DPA1*02:02, DPA1*02:03, and DPA1*04:01). The remaining genes had eplets commonly expressed across their alleles. It was noted that several alleles in the database did not encode for any antibody-verified eplets (DQA*01 alleles and DPB1*01:07). An interactive version of the networks can be accessed at www.gctransplant.ca/category/animations/.

A total of 361 unique HLA alleles were identified among the 1846 subjects, 206 of which were at HLA class I loci (59 at A, 107 at B, and 40 at C loci) and 155 at HLA class II loci (56 at DRB1, 7 at DRB3, 4 at DRB4, 6 at DRB5, 20 at DQA1, 18 at DQB1, 7 at DPA1 and 37 at DPB1 loci) (Table 1). The class I alleles individually encoded 0 to 11 eplets and class II alleles encoded 0 to 17 eplets. In total, the 361 alleles in the study population encompassed 150 eplets of which 59 were at class I loci (31 at A, 26 at B and 16 at C) and 91 were at class II loci (38 at DRB1/3/4/5, 11 at DQA1, 32 at DQB1, 2 at DPA1 and 8 at DPB1) (Table 1). The reduction in complexity observed when converting HLA alleles to eplets is depicted in Fig. 3a for class I and Fig. 3b for class II.

Numerous intra-locus eplets were identified, encoded by a range of multiple alleles within the same gene. The class I eplet 131S, for example, was encoded by 90 HLA-B alleles whereas the 163RG eplet was encoded by only 2 alleles (A*01:01 and A*01:02). This was also observed for class II, where 56A was encoded by 19 DPB1 alleles while the 25Q eplet was encoded by only one DRB1 allele. Multiple inter-locus eplets were also present, encoded by alleles in more than one gene (Supplementary Table 2). Thirteen

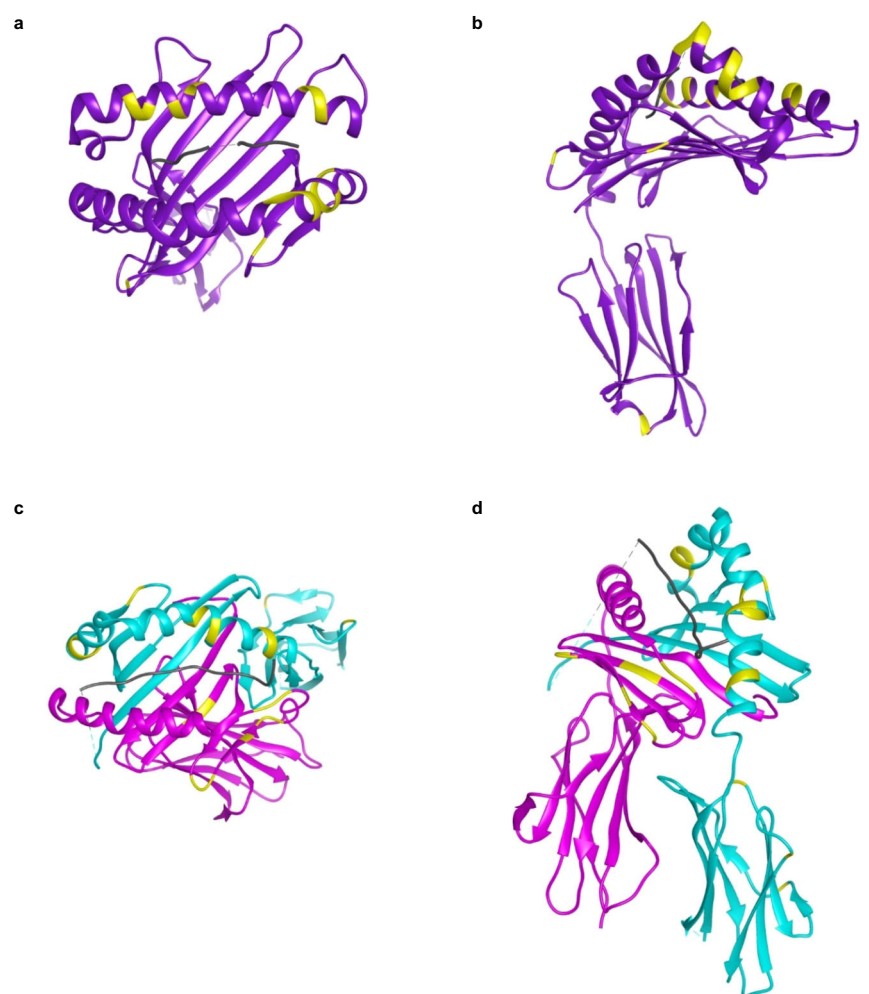

**Fig. 1 Class I and class II HLA proteins and their eplets. a** Top-down view of the peptide-binding groove of HLA-A*02:01 (without β2 microglobulin). The HLA-A*02:01 protein is shown in purple, a processed self-peptide in black, and its B-cell eplets highlighted in yellow. **b** Side-view of HLA-A*02:01, showing the alpha 3 portion and an eplet outside of the peptide-binding groove. **c** Top-down view of the peptide-binding groove of HLA-DQ. The DQA1*03:01 chain is in fuchsia, DQB1*02:01 is in aqua, a processed peptide in black, and eplets highlighted in yellow. **d** Side-view of HLA-DQ2.3 showing alpha 2 of the alpha and beta chains with highlighted eplets. Molecular graphics and analyses performed with UCSF Chimera, developed by the Resource for Biocomputing, Visualization, and Informatics at the University of California, San Francisco, with support from NIH P41-GM103311[13] using the Protein Data Bank ID: 4u6x (HLA-A*02:01)[14] and 4d8p (HLA-DQ2.3)[15].

eplets were encoded by two class I genes, and one eplet (163EW) occurred in all three class I genes. Fourteen eplets were encoded by more than one class II gene, restricted to the *DRB1/3/4/5* alleles. No eplets were shared between class I and class II alleles.

**Relative frequencies of HLA alleles and eplets**. Most of the 361 alleles observed occurred with low frequencies (Fig. 4a, b, Supplementary Data 1, Supplementary Fig. 2). Less than 2% ($n = 7$) were carried by more than 30% of subjects (class I: *A*02:01*; and class II: *DPA1*01:03*; *DPB1*04:01*; *DRB4*01:03*; *DQB1*03:01*; *DRB3*02:02*; and *DQA1*01:02*) while over half ($n = 188$) were present in less than 1% of subjects producing a highly skewed frequency distribution.

In contrast, eplets generally occurred with much higher frequencies (Fig. 4c, d, Supplementary Data 2, Supplementary Fig. 3). Of the 150 eplets identified, over three quarters ($n = 113$) were carried by more than 30% of subjects and the most frequent class I (79GT and 69TNT) and class II eplets (85VG(DRB), 25R, and 77T) were carried by 90% and 99% of subjects, respectively. Even the least frequent eplets (class II: 164VQ, 40D2 and class I: 17RS, 62LQ) were carried by 3–5% of subjects.

**Eplet frequencies are comparable in patients and donors**. Despite close linear correlation in allele frequencies between patients and donors ($r = 0.975$, Fig. 5a), some important differences were observed. For example, the most common class I alleles, *A*02:01*, occurred respectively in 31% of patients and 40% of donors while the most common class II allele *DPA1*01:03* occurred in 83% of patients and 91% of donors. Less common alleles were infrequent in both groups, for example, *B*42:01*, was present in 0.2% of patients and 0.1% of donors. Comparison of patient and donor sub-groups (Supplementary Fig. 4) showed close correlation between patients prior to or post-transplantation ($r = 0.992$) though greater disparity of both groups between deceased donors ($r = 0.949$, $r = 0.961$). Comparison of patients and living donors ($r = 0.973$, $r = 0.980$), and living and deceased donors ($r = 0.985$) produced similar results.

Eplet frequency profiles for donors and recipients were comparable across the frequency spectrum ($r = 0.991$) (Fig. 5b). The most common class I eplet, 79GT, encoded by all but 4 identified HLA-A alleles ($n = 48$), was present in 96% of patients and 97% of donors while the least common class I eplet, 17RS, encoded by 6 HLA-A alleles was present in 4% of patients and 3%

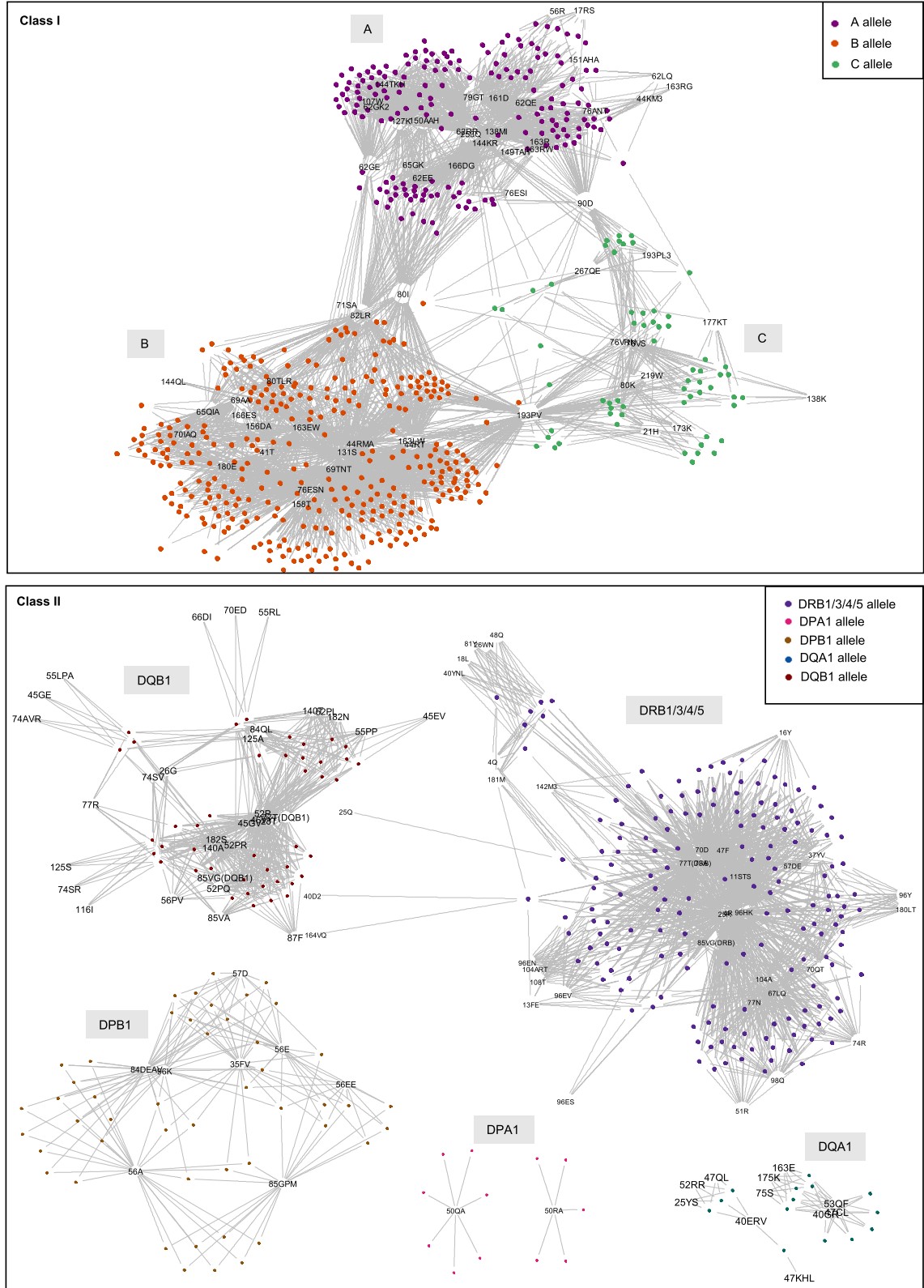

**Fig. 2 Networks[44] of the complete library of HLA alleles and their associated eplets present in HLAMatchmaker v02 for HLA class I genes and v02.2 for HLA class II genes.** Each node represents an allele and the lines connect them to eplets, showing that that allele expresses that eplet. Alleles are color-coded by gene. Interactive visualizations of the networks can be accessed at www.gctransplant.ca/category/animations/.

of donors. Similar results were observed for class II eplets where 85VG(DRB) occurred in 99% of both recipients and donors. Comparison of patient and donor sub-groups showed similar close correlation between patients prior to and post-transplantation with little disparity between both groups and deceased and living donors (Supplementary Fig. 4).

**Genotype frequencies**. A total of 1800 discrete genotypes, comprising the 16–18 alleles encoded at each of the 11 HLA gene loci on both chromosomes (note that *DRB3/4/5* may be absent or hemizygous in an individual genotype) were identified in the 1846 study subjects and combinations of these are shown in

Table 1. The 206 class I alleles identified were combined in 1572 discrete genotypes and the 155 class II alleles in 1509 discrete genotypes. Diversity at a single gene locus ranged from a maximum of 107 alleles and 602 genotypes at *HLA-B* to 7 alleles and 14 genotypes at *DPA1*.

Genotype distribution differed between patients and donors: 1017 complete genotypes (comprising all loci) were observed uniquely in patients and 756 uniquely in donors, with only 27 genotypes (1.5%) occurring in both groups (Fig. 6a). The number of shared genotypes increased as fewer gene loci were considered; for example, 6% of class I genotypes and 7.4% of class II genotypes were shared between patients and donors. The specific HLA gene locus was of primary importance: 30% of genotypes were shared at *DRB1/3/4/5* and *DQB1*, 37% at *DRB1/3/4/5*, 51% at *DPB1*, 64% at *DPA1*, and 78% at *DQA1*. Genotype sharing was most common at *DQB1*, with 90 genotypes (79%) occurring in both groups. No class I or class II genes had 100% of genotypes present in both patients and donors.

**Epitype frequencies**. A total of 1793 discrete epitypes, comprising the eplets expressed on the HLA proteins encoded by each of the 11 gene pairs (for example, the *A*30:01/A*33:01* genotype encodes the epitype: 17RS, 56R, 62RR, 62QE, 79GT, 138MI, 253Q) were identified in the 1846 subjects, with frequencies ranging from a maximum of 4 (0.22%) to a minimum of 1 (0.05%) (Table 1). The number of identified epitypes was marginally lower than that of genotypes at all individual loci and locus combinations. For example, the 59 HLA class I eplets identified were combined in 1487 discrete epitypes and the 91 class II eplets in 1086 discrete epitypes. Diversity at a single gene locus ranged from a maximum of 26 eplets and 288 epitypes at HLA-B, to 2 eplets and 3 epitypes at DPA1.

Epitype distribution also differed between patients and donors: 1010 epitypes (comprising all loci) were observed uniquely in patients and 751 uniquely in donors, with only 32 epitypes (1.8%)

| | Alleles | Genotypes | Eplets | Epitypes |
|---|---|---|---|---|
| **Table 1 Summary of HLA alleles, genotypes, eplets, and epitypes in the total study population of 1846 subjects at various loci combinations and individual loci.** | | | | |
| All Loci | 361 | 1800 | 150 | 1793 |
| Class I | 206 | 1572 | 59 | 1487 |
| Class II | 155 | 1509 | 91 | 1086 |
| DRB1/3/4/5+DQB1 | 91 | 710 | 70 | 365 |
| A | 59 | 253 | 31 | 128 |
| B | 107 | 602 | 26 | 288 |
| C | 40 | 227 | 16 | 74 |
| DPA1 | 7 | 14 | 2 | 3 |
| DPB1 | 37 | 146 | 8 | 24 |
| DQA1 | 20 | 109 | 11 | 10 |
| DQB1 | 18 | 115 | 32 | 34 |
| DRB1 | 56 | 404 | 28 | 161 |
| DRB3 | 7 | 14 | 10 | 7 |
| DRB4 | 4 | 8 | 9 | 3 |
| DRB5 | 6 | 13 | 11 | 4 |
| DRB1/3/4/5 | 73 | 564 | 38 | 199 |

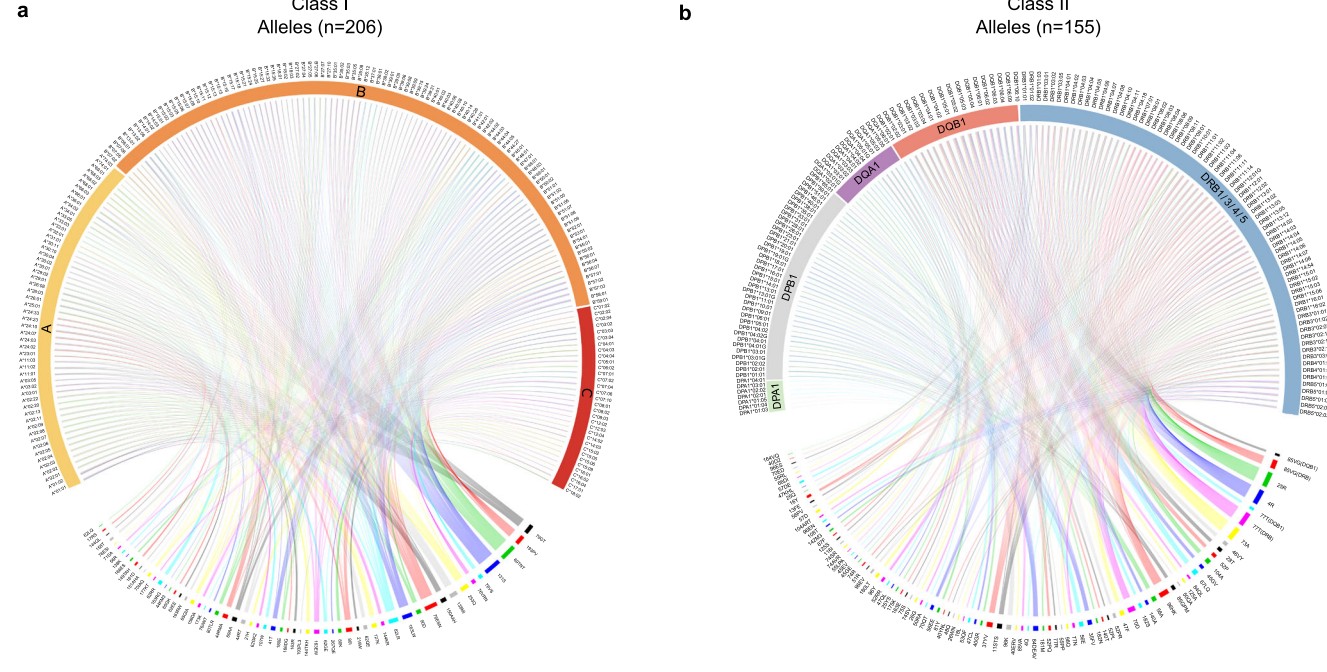

**Fig. 3 The reduction of HLA complexity of identified alleles in the study population and their conversion into eplets[45,46].** Conversion of alleles to eplets was determined by HLAMatchmaker. HLA alleles identified in the study population are shown in top portion and eplets shown in the bottom portion. Interconnections represent an allele encoding an eplet or vice versa. **a** Class I alleles and eplets and **b** Class II alleles and eplets.

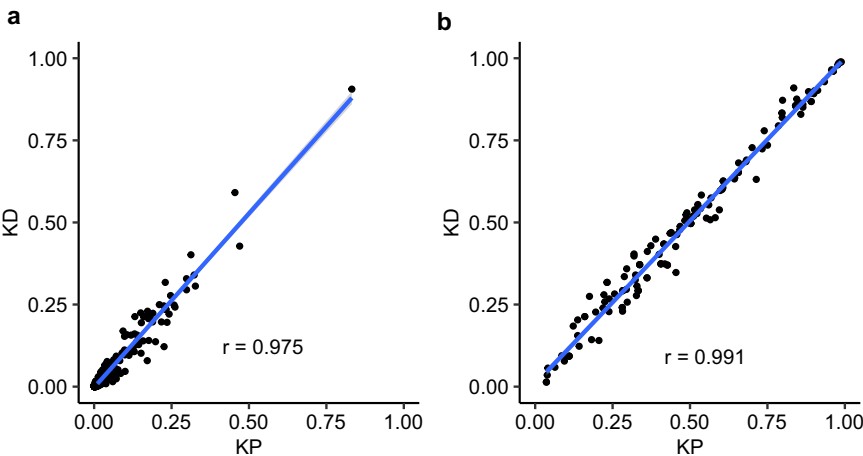

**Fig. 4 The relative frequencies of HLA alleles and eplets in the study population.** Relative frequencies were calculated as the proportion of subjects expressing a particular allele or eplet within kidney patients ($n = 1049$) or donors ($n = 797$). **a**, **b** Depict the allele frequencies by class I and II, respectively. **c**, **d** Depict eplet frequencies by class I and II, respectively. KP kidney patients, KD kidney donors.

**Fig. 5 Pairwise analysis between patient and donor HLA allele and eplet frequencies.** Relative frequencies were calculated as the proportion of subjects expressing a particular allele or eplet. The line and correlation coefficient were calculated using Pearson correlation. **a** Each dot represents an allele and its frequency for patients ($n = 1049$) is plotted on the x axis and its frequency for donors ($n = 797$) is plotted on the y axis. **b** As above but for eplets. KP kidney patients, KD kidney donors.

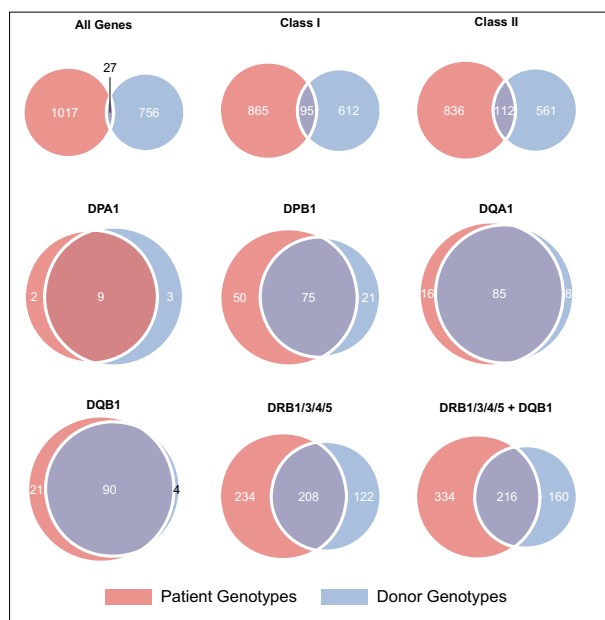

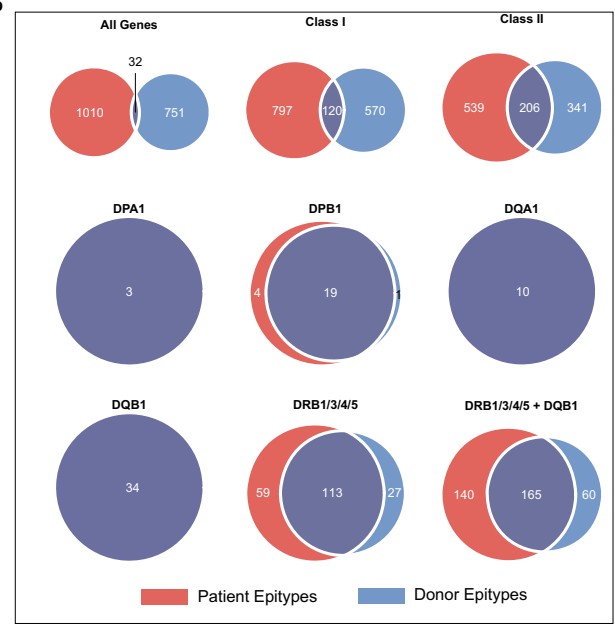

**Fig. 6 Genotype and epitype frequencies between patients and donors. a** Venn diagrams of unique genotypes and how some or all occur between patients (n = 1049) and donors (n = 797). Genotypes were determined at the individual level, where alleles of a particular loci were sorted and combined. Analysis was performed on various loci combinations: all 11 HLA genes, HLA class I, HLA class II, *DPA1, DPB1, DQA1, DQB1, DRB1/3/4/5,* and *DQB1* and *DRB1/3/4/5* combined. **b** As above but for epitypes.

being observed in both groups (Fig. 6b). The number of shared epitypes increased as fewer gene loci were considered; for example, 8% of class I epitypes and 19% of class II epitypes were shared between patients and donors. The specific HLA gene locus was also important: 45% of epitypes were shared at *DRB1/3/4/5* and *DQB1*, 57% at *DRB1/3/4/5*, and 79% at *DPB1*. For *DPA1, DQA1,* and *DQB1*, 100% of epitypes occurred in patients and donors, with a significant decrease in the number of epitypes compared to genotypes.

**Eplet mismatch distribution.** Base-case modeling estimated the probability of patient–donor eplet mismatch by comparing all deceased donors and patients, generating a total of 92,756 potential matches. The numbers of eplet mismatches across all genes and various gene combinations are shown in Fig. 7a. Median mismatch (and ranges) were 27 (0–65) for all 11 genes, 10 (0–27) for class I genes, 17 (0–46) for class II genes, 6 (0–21) for *DQB1* only, 6 (0–20) for *DRB1/3/4/5* genes, and 12 (0–39) for *DRB1/3/4/5* and *DQB1* genes. The probability of a 0 eplet mismatch by chance alone was 12% at *DQB1*, 6% at *DRB1/3/4/5*, and <5% for all other combinations.

Eplet mismatch frequencies were inversely correlated with study population frequencies ($r = -0.998$) (Fig. 7b) in that common eplets were infrequently mismatched and vice versa. For example, the most common eplet 85VG(DRB) (present in 99% of subjects) occurred in 1.3% of possible mismatches. In comparison, 40D2 occurred in 2.7% of subjects but occurred in 97% of possible mismatches.

Furthermore, the mismatch results showed a high probability of identifying a recipient for each successive donor with a mismatch score of 0 at *DRB1/3/4/5*, at *DQB1* and at these combined gene loci. For *DRB1/3/4/5*, 93% of donors matched at least one patient with a mismatch score of 0, while a further 5% had a minimum mismatch score of 1, and 0.8% a score of 2. For *DQB1*, all donors matched at least one patient with a mismatch score of 0 except for one donor with a minimum mismatch score of 1. For

the combined *DRB1/3/4/5* and *DQB1* loci, 84% of donors could be matched with a patient having a mismatch score of 0 at all 5 loci, with a 95% having a mismatch score of score 1 or less, and 97.5% a score 3 or less.

**Organ allocation simulations with prospective eplet matching.** Simulation was performed using the Canadian national data comprising 9 provincial programs with a combined waiting list of 2032 patients and 762 deceased donors (~1500 kidneys) per year (Fig. 8). Prospective matching within this national pool enabled a high degree of eplet match, with full compatibility at the critical *HLA-DR* and *DQ* loci. Compared with no matching, the median mismatch score (and range) declined from 27.35 (0–62) to 9.3 (0–22) for the full epitype; for class I from 10.2 (0–27) to 3 (0–11) and for class II from 16.8 (0–45) to 1 (0–13); and for *DRB1/3/4/5* from 6 (0–20) to 0 (0–4), for *DQB1* from 6 (0–21) to 0 (0–5), and for *DRB1/3/4/5* and *DQB1* combined from 12.3 (0–39) to 0 (0–10).

Modeling the average provincial program (waiting list 290 patients; 109 deceased donors/year) also predicted a high degree of eplet match (Supplementary Fig. 5) with full compatibility at the critical *HLA-DR* and *DQ* loci. Median mismatch score (and range) declined from 27.15 (0–60) to 11.8 (0–31) for the full epitype; for class I from 10.25 (0–24) to 3.9 (0–13) and for class II from 16.8 (0–42) to 3.3 (0–27); and for *DRB1/3/4/5* from 5.95 (0–19) to 0.1 (0–11), for *DQB1* from 5.9 (0–21) to 0 (0–11), and for *DRB1/3/4/5* and *DQB1* combined from 12.2 (0–36) to 0 (0–21).

Scenario analyses performed across the range of provincial programs confirmed the importance of waiting list size in determining matching success, with an inflexion at approximately 250 recipients indicating the minimum number to optimize eplet matching (Fig. 9a). The probability of perfect eplet identity (mismatch score = 0) at *DQB1* was 92% with a waitlist of 790 and compared with 65% with a waitlist of 88. Requirement for good, as opposed to perfect identity (a cumulative mismatch score of 10 or lower), improved the probability of successful matching across

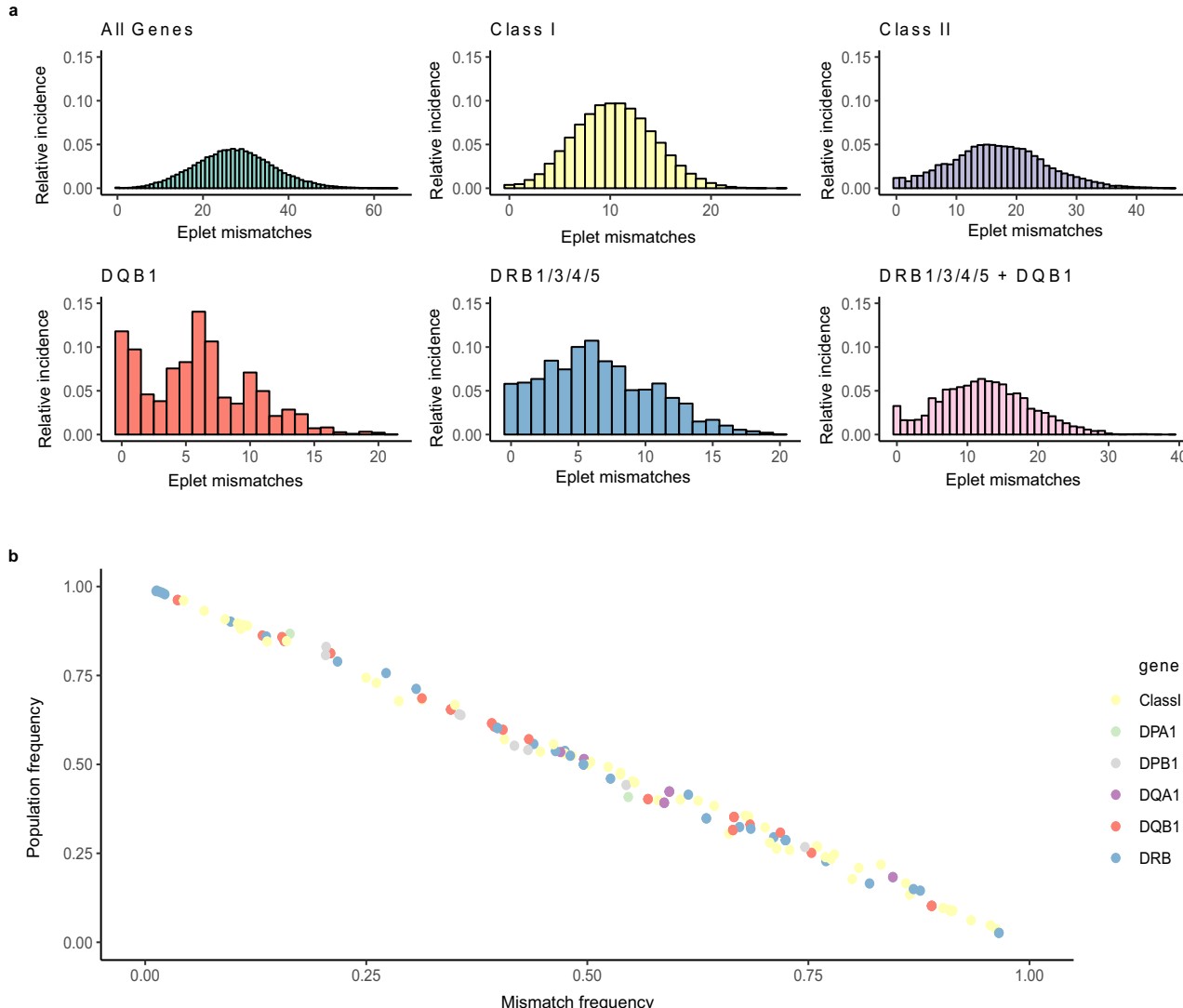

**Fig. 7 The eplet mismatch distribution in the study population.** 1049 kidney patients were matched against 243 deceased donors by blood-group identity. Next, eplet mismatches across all 11 HLA genes (*HLA-A, -B, -C, -DRB1, -DRB3, - DRB4, -DRB5, -DPA1, -DPB1, -DQA1,* and *-DQB1*), class I genes, class II genes, *DQB1* only, *DRB1/3/4/5*, and *DRB1/3/4/5* and *DQB1* were calculated. A mismatch was determined as an eplet present in a donor that is not present in the patient. **a** Distribution of the relative incidences of eplet mismatch scores at all gene combinations analyzed. Relative incidence is the proportion of a particular quantitative eplet mismatch out of the total number of 92,756 blood-group identical matches. **b** The linear correlation (Pearson's) of population frequency versus mismatch frequency for individual eplets. A dot represents an individual eplet color-coded by gene combination, and all identified eplets are plotted (*n* = 150). Population frequency is the proportion of individuals (kidney patients and donors) whom express the eplet, out of the total study population (*n* = 1846). Mismatch frequency was the incidence of a particular eplet being mismatched in blood-group identical matches, divided by the sum of the number of donors with the specific eplet multiplied by the number of patients, restricted by blood group. The correlation coefficient (*r*) is −0.998.

all programs and gene loci except for the full epitype (Fig. 9b), with even the smallest program achieving 87% successfully matched pairs. See Supplementary Data 3 for the cumulative probabilities of eplet mismatch scores in all simulations with prospective eplet matching.

Extension of these models to other organs showed that regional or national sharing may be required to enable epitope compatibility for heart, lung, and liver transplants whose national waiting lists (*n* = 72, 150, and 285) and donor totals (*n* = 141, 306, and 430) are smaller than those of the kidney[3].

## Discussion
Complementary genomic and proteomic methods have clarified the structural biology of HLA antigens, enabling more precise understanding of the complex direct, indirect, and semi-direct mechanisms of allorecognition by recipient lymphocytes[16–19]. Two cardinal groups of epitopes are now recognized, those involved in indirect recognition of the donor HLA antigen array by recipient T cell which are predicted through the PIRCHE algorithm[20], and those which are antibody-accessible and are involved in the humoral response (defined here as B-cell epitopes), predicted through the HLAMatchmaker algorithm[10]. Studies confirm that the mismatch between donor and recipient for each of these two sets of molecular targets is directly related to the risk of rejection and graft loss[11,12,21–23]. In this study we focus on antibody-accessible eplets, restricting our attention to those for which biological relevance has been verified by the detection of specific antibodies to the target (http://www.epitopes.net/publications.html). The majority of these are encoded within exons 2 and 3 for class I and exon 2 for class II antigens, though

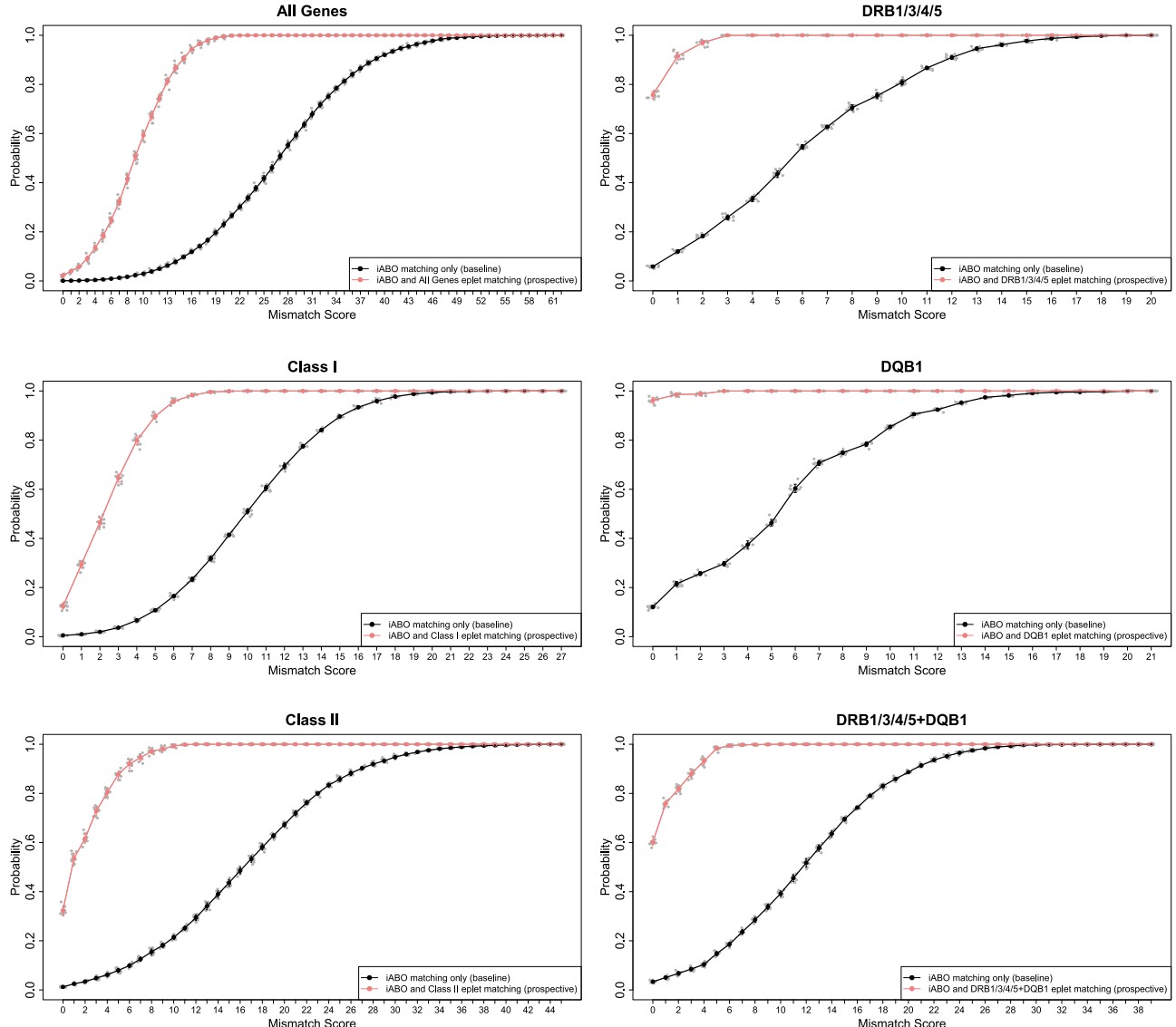

**Fig. 8 Cumulative probabilities of eplet mismatches in the national matching simulations with and without prospective eplet matching.** Matching simulations incorporated prospective eplet and blood group matching (red curve) and baseline blood group matching only (black curve) in kidney patients and deceased donors. Plots represent simulations using a national Canadian waitlist ($n = 2032$ patients) and an annual deceased donor rate ($n = 762$ donors) according to Canadian Organ Replacement Register[3]. Simulations were performed across all HLA genes, class I genes, class II genes, DRB1/3/4/5, DQB1, and DRB1/3/4/5 + DQB1 combined. Eplet mismatch scores for the respective genes on the $x$ axis are plotted against the averaged cumulative probability of these scores in the matched population. Jittered grey dots represent the cumulative probabilities at each individual simulation. Error bars are calculated as the standard deviation of ten repeated independent simulation runs.

eplets can exist outside these regions on the expressed protein. We document the probability distributions of these eplets and the broader epitopes determined unambiguously by NGS among patients and donors in a large and ethnically diverse transplant program, and estimate the quantitative mismatches achievable at each HLA gene locus with or without prospective matching. These data provide the basis to inform strategic decisions for incorporating quantitative epitope mismatch data into clinical practice, either through retrospective use of a mismatch score to estimate risk and adjust immune suppression, or by prospectively matching donors and recipients to minimize incompatibility and improve overall outcomes.

The results reported reinforce the small proportion of documented HLA alleles commonly observed in routine practice[24]. Only 361 of the more than 25,000 class I and class II HLA alleles (1–2%) were observed in the patients and donors from this highly

ethnically diverse transplant population, most of which were present in fewer than 5% of subjects; only 7 individual alleles were observed in more than 30% of cases. Matching for identity at the allele level is therefore challenging. Large donor registries have been established to achieve this in hematopoietic stem cell transplantation, a strategy not feasible in organ transplantation[25]. Eplets, in contrast, are fewer in number and often distributed across multiple alleles within or between gene loci, resulting in higher frequencies and more linear distribution. Of the 150 eplets identified, three quarters were present in more than 30% of subjects and several occurred in over 90% of subjects in both patient and donor groups, increasing the potential for prospective matching to enhance compatibility.

Transplant patients and deceased donors often differ in ethnicity especially in an ethnically diverse population such as British Columbia[26], raising concern that allelic diversity may

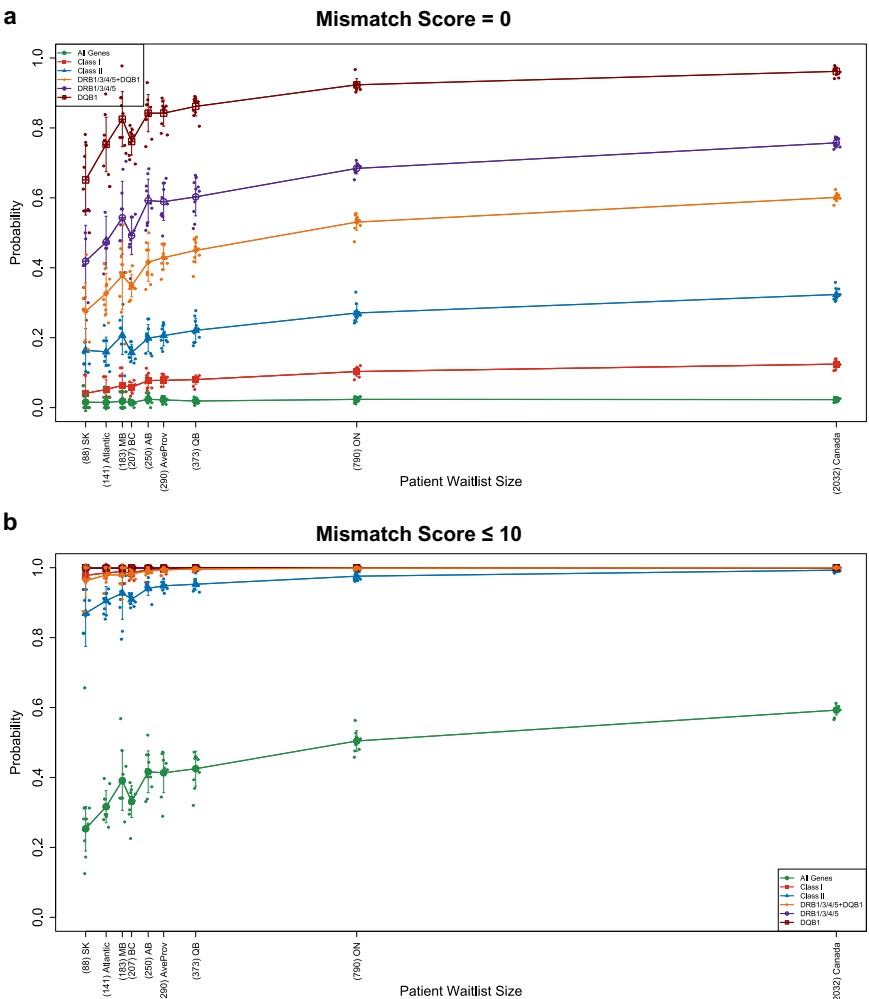

**Fig. 9 The effect of patient waitlist size on mismatch score in prospective eplet matching simulations.** Provincial active waitlist and deceased donor numbers were used according to Canadian Organ Replacement Register[3]. *X* axis shows the number of patients on provincial or national waiting lists (SK Saskatchewan with 88 patients, Atlantic Atlantic provinces (New Brunswick, Nova Scotia, and Prince Edward Island) with 141 patients, MB Manitoba with 183 patients, BC British Columbia with 207 patients, AB Alberta with 250 patients, AveProv Average across all provinces with 290 patients, QC Quebec with 373 patients, ON Ontario with 790 patients, and Canada with 2,032 patients). *Y* axis shows the averaged cumulative probability of achieving a total eplet mismatch score of 0 (**a**) or 10 or lower (**b**). Jittered dots represent the cumulative probabilities at each individual simulation. Error bars are calculated as the standard deviation of ten repeated independent simulation runs.

create an inequality barrier by increasing waitlist times for ethnic minorities with rare genotypes, diminishing the enthusiasm for stringent HLA matching for deceased donor transplantation in Canada and other locations of high population diversity[27–29]. We observed differences in allele carrier rates between deceased donors and patients in this study, while both patient sub-groups, prior to or post-transplantation serving as internal controls, showed tight correlation with each other ($r = 0.992$). Conversion from alleles to eplets not only reduced the HLA complexity but minimized the consequences of ethnic diversity. Every eplet was present in both patients and donors, compared with 29% of alleles that were absent in one or other of these groups. In consequence, when examined at the epitope level, correlation was robust between donors and patients prior to ($r = 0.988$) or post-transplant ($r = 0.980$), comparable to the correlation between these two individual patient groups ($r = 0.996$).

Although eplets are widely expressed in donors and patients, often overlapping in clusters related to the presence of common class I or class II alleles, the number of discrete epitopes (eplets present at all 11 gene loci) identified ($n = 1793$) was very similar to the number of HLA genotypes ($n = 1800$). No more than 1–2%

of epitypes were shared between donors and patients, with only 6% sharing at the class I region and 7.4% at the class II region, indicating that identical matching at the epitope or at each gene region is improbable in a diverse transplant population. However, increasing data suggest that eplet compatibility at certain class II gene loci, particularly *HLA-DRB1/3/4/5* and -*DQB1* is of primary importance in minimizing graft injury[11,12,23], reflecting the frequent occurrence of antibodies to these gene products in AMR[30–32]. Our studies utilizing the US Scientific Registry of Transplant Recipients (SRTR) confirm this understanding, documenting an increased risk of transplant glomerulopathy and death-censored graft failure in donor–recipient pairs mismatched at these loci[33]. The data reported here demonstrate that the probability of identity at these class II loci is substantially higher than for the full epitope, ranging from 30% at *HLA-DRB1/3/4/5* and -*DQB1* to 79% at *DQB1* alone, so providing a logistical basis for deliberate matching at these loci.

In focusing on AMR, we have primarily examined antibody-verified eplets and employed mismatch counts as a measure of incompatibility. However, since our understanding of relative immunogenicity remains limited, it is likely that not all

polymorphisms have the same biological effect in the rejection process and that structural properties of the eplet could be equally or more important than the simple quantity of mismatches. This raises critical questions related both to the specific eplet differences and the quantitative sum of these differences between recipient and donor. For example, a higher mismatch score may confer a summative biological effect, or simply increase the probability of a highly immunogenic eplet being present in the donor. The immunogenicity of an eplet is relevant only in the context of the three-dimensional structure of the HLA proteins expressed by the transplant recipient. Kosmoliapsis et al. also consider the disparity of physiochemical properties (i.e., electrostatic score, EMS) of mismatched amino acids, and report a greater EMS to be an independent predictor of the formation of antibodies to donor *DR* and *DQ* targets[34]. The location of the mismatched eplet may also play a role in its immunogenicity as described by Tambur[35]. Polymorphisms occur at different locations in the HLA protein, including both the peptide-binding cleft and the outer aspects of the molecule, which influence how peptide is presented to the T cell. More precise understanding of eplets and their role in transplant rejection is therefore critical to refine matching algorithms.

The use of retrospective mismatch scores to select patients at risk of rejection and to modify management is valuable, but does not maximize the benefit of eplet compatibility. As shown here, the allocation of deceased donors to transplant patients without deliberate HLA matching showed a median eplet mismatch of 27 across the full epitype, 10 across class I genes, and 17 across class II genes. Even at individual genes, the estimated median mismatch score ranged from 6 (range: 0–21) at *DQB1* alone to 12 (range: 0–39) at *DRB1/3/4/5* and *DQB1*, suggesting a wide range of rejection risk in non-matched transplant populations. But the implementation of prospective matching requires thoughtful consideration of logistics. Here we show that the opportunity to maximize compatibility by prospective eplet matching is influenced by the size of the transplant waitlist. Simulation modeling based on a broad range of donor and patient frequencies indicated that a waitlist of 250 or more active recipients offers the greatest opportunity to achieve optimal matching at key class II HLA gene loci. This number, which is consistent with the waiting list in major transplant centres in America, Europe, and many other countries, is important in informing organ sharing policies within regions, particularly in geographic areas with low population densities, since the logistics, costs and storage time involved in organ sharing must be balanced against the benefits achieved in prolonging outcomes[36–39].

These results provide compelling evidence that prospective donor/recipient eplet matching is feasible in the Canadian population and, while not enabling full epitype identity, matching may successfully achieve a very low—or zero—eplet mismatch at the critical HLA class II loci in the majority of patients awaiting transplant. Further, the results indicate that a high degree of successful matching at these loci may be achieved within program or region, assuming a minimum waiting list of 250 patients. This is of vital importance since, while national organ sharing is routine for highly-sensitized patients, the costs and logistical complexities of transporting all organs nationally would be substantial. Our data suggest that, in certain cases, a small number of contiguous programs may need to be combined to ensure transplant regions with adequate waitlist numbers for kidney, and that regional or national sharing will be required for non-renal organs. But graft success or economic costs are not the sole arbiters of policy and the utility donated organs must be balanced by equality of access to them[27]. Matching at the eplet level may more closely approximate this latter goal than the simple use of allele compatibility, although accommodation must still be considered for recipients with uncommon eplets of high biological importance. Potential accommodations include giving priority to these waitlisted patients (i.e., much like the Highly Sensitized Patients (HSP) Program), extending their minimum mismatch criteria, and setting a maximum time on the list until transplantation given all other criteria are met.

Limited approaches to eplet matching have been incorporated by other programs. Eurotransplant has successfully used class I eplet to define acceptable mismatches and expand the donor pool for highly-sensitized patients[40–42] and eplet matching of class I (<10 eplets) and class II (<30 eplets) has been performed in Australian pediatric patients[43]. Although these examples are small the results show promise.

This study has certain limitations which we are working to address. We restricted this analysis to antibody-verified eplets because of their demonstrated clinical importance in AMR. It is possible that putative B-cell eplets for which antibodies have not yet been identified, or other eplets which are recognized primarily by the T-cell receptor (PIRCHE), also play an important role in graft rejection and we are, therefore, evaluating these in subsequent analyses. A small number of subjects (8%) could not be included since they expressed one or more alleles that are not present in HLAMatchmaker. We are working to update HLA-Matchmaker to include these alleles and will re-evaluate once this process is complete. Subject selection and sequencing were performed in a single provincial program, raising potential concern for both precision and representativeness. B.C. has served as the lead program for evaluation, validation, and implementation of NGS for HLA genes in Canada, and the accuracy and reproducibility of these assays have been fully validated according to American Society of Histocompatibility and Immunogenetics (ASHI) standards. But the ethnic diversity of the program and the small number of subjects excluded, we believe the data is highly representative of the eplet frequencies observed across Canada. We are currently engaged in a larger study to confirm these national data. In modeling matching probabilities, we extrapolated the allelic and eplet frequencies observed in this program to the broader pool of patients and donors across Canada. Precise national population frequencies will enable us to refine these model probabilities. And we acknowledge the limitations of model parameters, which we have deliberately restricted to allocation by blood group identity and optimal eplet match. Recognizing these considerations, we present the first data describing HLA eplet frequencies in patients and donors in a highly diverse ethnic population. We show that the conversion of alleles to eplets reduces the HLA complexity and enables matching at selected clinically important HLA genes. And in a simple allocation model, we demonstrate that a high degree of eplet compatibility can be achieved at these loci with a relatively small waiting list, reducing the requirement for national transport of all donor organs and so minimizing costs and ex-vivo storage time. We are now proceeding with studies to define more precisely the immunogenicity of dominant eplets and to incorporate eplet frequencies from other regions in a more comprehensive model to guide allocation within a national program.

## Methods

This population-based study included renal transplant patients and donors genotyped by next-generation sequencing (NGS) from October 2016 to January 2019 at the Provincial Reference Immunology Laboratory (Vancouver General Hospital, Vancouver, BC). This research was approved by the University of British Columbia Clinical Research Ethics Board (#H18-00090).

**HLA gene sequencing, eplet analysis, and carrier frequencies**. DNA was extracted from whole blood using the EZ1 DNA Blood 350 μl Kit or QIAsymphony DSP DNA Mini Kit (192) (Qiagen, Hilden, Germany). Whole gene characterization

(5′UTR to 3′UTR) for HLA-A, B, C, DQA1, DPA1, and DQB1 and key region characterization (exon 2 to 3′UTR) of HLA-DRB1, DRB3, DRB4, DRB5, and DPB1 was performed using the Holotype HLA kit version 2 (Omixon, Budapest, Hungary). Libraries were sequenced using MiSeq Sequencer (Illumina, California, USA) and sequence data were analyzed using the HLA Twin version 2.5 (Omixon, Budapest, Hungary).

HLA eplets were obtained from allelic data using the computer algorithm HLAMatchmaker v02 for HLA class I and v02.2 for HLA class II genes (www. epitopes.net) which uses a database of alleles to define a string of eplets for each allele. Only eplets registered to be experimentally confirmed by antibody testing (antibody-verified eplets) were used in this analysis.

Allele or eplet carrier rates (relative frequencies) were calculated respectively as the proportion of subjects expressing a specific allele or eplet within the total study sample. Composite genotypes (HLA types) and their corresponding epitypes were calculated respectively as the numbers of subjects within the sample expressing each unique combination of alleles or eplets at all the 11 allelic HLA gene loci.

**Eplet matching and simulation.** An allocation simulation framework was implemented in R (MRAN 3.5.2 and 3.5.3) to model kidney matching between deceased donors and patients. Simulation was initialized using wait lists of specified size and bootstrapped recipients were added one at a time until the determined number was reached. This process produced an initial rank ordering where the first recipient added was at the top, and the last recipient added was at the bottom of the wait list. Each donor was considered to provide two donor kidneys for matching with recipients on the wait list using defined matching rules (see below). Before the next donor was entered into the model, two new recipients were randomly selected from the recipient distribution and added to the bottom of the wait list, keeping the wait list at a constant size over the course of the simulation. Simulation continued until all donor kidneys were allocated. Eplet mismatch was defined as the presence of an eplet in a donor that was not present in the recipient. Numerical eplet mismatches were calculated across all genes or selected loci combined for each donor–recipient pair. Match-information, including eplet and epitype mismatch scores, were recorded and stored in a Match-List table for post-simulation analysis.

The baseline scenario was structured to approximate the current Canadian allocation model in which deceased donor allocation is performed primarily within the province of kidney origin, with national sharing for a small proportion of primarily highly-sensitized patients. Time on the wait list is the principal determinant of ranking order, adjusted to allow for clinical priority in a small proportion of subjects (e.g., children, loss of dialysis access, other cases of exceptional clinical urgency), and constrained by ABO identity to avoid over-allocation of group O donors to non-O recipients. Within this system, HLA compatibility is used only as a lower-level decision factor to select between individuals of otherwise equal ranking, and organs are therefore normally allocated independent of eplet mismatch.

Exploratory simulation models were then developed to examine the impact of deliberate eplet matching across a range of wait-list and donor pool sizes representing Canadian provinces obtained from the 2018 Canadian Institute of Health Information data (https://www.cihi.ca/en/organ-replacement-in-canada-corr-annual-statistics-2019). Allocation was modeled with constraints for ABO identity, and organs were assumed to be freely shared across Canada in the full national model. Eplet mismatching was calculated for all HLA 11 genes, for class I and class II regions, and for DRB1/3/4/5/+DQB1, DRB1/3/4/5 and DQB1 loci. Each donor was considered to be matched with the recipient having the lowest eplet mismatch score at the relevant HLA locus or loci, with the rank on the waitlist determining priority in cases of identical scores. Sixty-three sets of simulations were performed with 10 replicates each (i.e., running the same simulation for different random orderings of recipients and donors) and the cumulative probability of increasing mismatch scores was derived for each scenario over the 10 replicates.

**Statistics and reproducibility.** The dataset was summarized providing the $n$ value overall and for each group, with mean, median, range, and standard deviations for continuous variable and counts and proportions for categorical variables. Correlations between allele and eplet frequencies were determined using Pearson's correlation coefficient ($r$). Simulations were conducted as described in "Eplet matching and simulation".

**Reporting summary.** Further information on research design is available in the Nature Research Reporting Summary linked to this article.

## Data availability
For reasons of clinical confidentiality, the dataset of the B.C. provincial transplant program employed in this article comprising of the HLA genotypes of the patients and donors has not been posted to a public site but may be accessed through the corresponding author under a formal data sharing agreement.

## Code availability
Custom code in R (based on MRAN 3.5.2 and 3.5.3) was used to analyze the data through the corresponding author under a formal data sharing agreement.

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

## Acknowledgements

We are indebted to the patients and donors whose samples were used in this study. We thank Evelyn Devera, Jennifer Beckrud, Brendan McKenzie, Edlyn Gomez, Tony Yeung, and the remaining members of the BC Provincial Immunology Laboratory for their assistance in genotyping and data retrieval. We are thankful to the members of the Genome Canada Transplant Consortium for their contribution to this research. Research is supported by Genome Canada, Genome British Columbia, Genome Quebec, Genome Alberta, Canadian Institutes of Health Research, and funded by awards LSARP 273AMR and GP1-155871 and partnered grants from Omixon.

## Author contributions

J.N.T., O.P.G., K.R.S., and P.A.K. contributed to the conception and the design of the study and wrote the manuscript. J.N.T. performed the sequencing, collection of data, and analysis of alleles and eplets and their frequencies. O.P.G. performed the network analysis and simulations. P.A.K. guided the research and provided oversight of the work. J.N.T., O.P.G., P.A.K., K.R.S., F.F., L.L.A., J.L., R.S.-P., F.C., S.G.E.M., R.M., and R.D. contributed to the review and interpretation of the data or results. All authors contributed to the drafting and critical review of the article and provided final approval of the manuscript.

## Competing interests

The authors declare no competing interests.

## Additional information

## Genome Canada Transplant Consortium

**Canada Working Group** Paul A. Keown [1,3,9✉], Ruth Sapir-Pichhadze[4], Stirling Bryan[10], Timothy Caulfield[11], Ioannis Ragoussis[12], Karim Oualkacha[13], Kathryn Tinckam[14,15], Robert Liwski[16], Patricia Campbell[17], Heloise Cardinal[18], Sacha A. De Serres[19], Lenka L. Allan[1], Chee Loong Saw[20], Michael Mengel[21], Banu Sis[21], Karen R. Sherwood[1], Eric Wagner[22], Noureddine Berka[23], Bruce McManus[1], W. Robert McMaster[8,9], Marie-Josée Hebert[24], Leonard J. Foster[25], Fabio Rossi[26], Christoph H. Borchers[27,28,29], Ciriaco A. Piccirillo[30,31], Constantin Polychronakos[32], Raymond Ng[33], Anthony Jevnikar[34], Pieter Cullis[25], Guido Filler[35], Harvey Wong[36], Bethany Foster[31,32], John Gill[3], S. Joseph Kim[14,15], Lee Anne Tibbles[37], Atul Humar[14], James Lan[1,3], Steven Shechter[38], Prosanto Chaudhury[39], Nicolas Fernandez[40], Elizabeth Fowler[41], Bryce Kiberd[42], Jagbir Gill[43], Marie-Chantal Fortin[18], Scott Klarenbach[44], Robert Balshaw[45], Seema Mital[46],

Istvan Mucsi[14,15,47,48], David Ostrow[3], Calvin Stiller[49], Rulan S. Parekh[14,15,46], Lucie Richard[50], Lynne Senecal[51] & Tom Blydt-Hansen[52]

**United States Working Group** Rene Duquesnoy[5], Henry Erlich[53], Howard Gebel[54], Eric Weimer[55], Bruce Kaplan[56] & Gilbert Burckart[57]

**United Kingdom Working Group** Derek Middleton[58] & Steven G. E. Marsh[7]

**Netherlands Working Group** Marcel Tilanus[59], Frans Claas[6] & Teun van Gelder[60]

**Germany Working Group** Gerhard Opelz[61] & Michael Oellerich[62]

**France Working Group** Pierre Marquet[63]

**New Zealand Working Group** Carlo Marra[64]

**Hungary Working Group** Zoltán Kaló[65]

[10]School of Population and Public Health, University of British Columbia, Vancouver, BC, Canada. [11]Faculty of Law and School of Public Health, University of Alberta, Edmonton, AB, Canada. [12]Department of Human Genetics, McGill University, Montreal, QC, Canada. [13]Department of Mathematics, University du Quebec à Montreal, Montreal, QC, Canada. [14]Department of Medicine, University of Toronto, Toronto, ON, Canada. [15]Division of Nephrology, Toronto General Hospital, University Health Network and University of Toronto, Toronto, ON, Canada. [16]Department of Pathology, Dalhousie University, Halifax, NS, Canada. [17]Department of Medicine and Dentistry, University of Alberta, Edmonton, AB, Canada. [18]Department of Medicine, University of Montreal, Montreal, QC, Canada. [19]Department of Medicine, Laval University, Quebec City, QC, Canada. [20]HLA Laboratory, Hematology Division, McGill University Health Centre, Montreal, QC, Canada. [21]Department of Laboratory Medicine & Pathology, University of Alberta, Edmonton, AB, Canada. [22]Department of Microbiology, Infectious Diseases, and Immunology, Laval University, Quebec City, QC, Canada. [23]Department of Pathology, University of Calgary, Calgary, AB, Canada. [24]Research Centre, Université de Montréal, Montreal, QC, Canada. [25]Department of Biochemistry and Molecular Biology, University of British Columbia, Vancouver, BC, Canada. [26]School of Biomedical Engineering, University of British Columbia, Vancouver, BC, Canada. [27]Department of Oncology, McGill University, Montreal, QC, Canada. [28]Segal Cancer Proteomics Centre, McGill University, Montreal, Canada. [29]Center for Computational and Data-Intensive Science and Engineering, Skolkovo Institute of Science and Technology, Moscow, Russia. [30]Department of Microbiology and Immunology, McGill University, Montreal, QC, Canada. [31]Research Institute of McGill Health Center, Montreal, QC, Canada. [32]Department of Pediatrics, McGill University, Montreal, QC, Canada. [33]Department of Computer Science, University of British Columbia, Vancouver, BC, Canada. [34]Department of Microbiology and Immunology, Western University, London, ON, Canada. [35]Department of Pediatrics, Western University, London, ON, Canada. [36]Faculty of Pharmaceutical Sciences, University of British Columbia, Vancouver, BC, Canada. [37]Cumming School of Medicine, Division of Nephrology, University of Calgary, Calgary, AB, Canada. [38]Sauder School of Business, University of British Columbia, Vancouver, BC, Canada. [39]McGill University Health Centre & Research Institute, Montreal, QC, Canada. [40]Université de Montréal, Montreal, QC, Canada. [41]Kidney Foundation of Canada, Burnaby, BC, Canada. [42]Division of Nephrology, Dalhousie University, Halifax, NS, Canada. [43]Division of Nephrology, University of British Columbia, Vancouver, BC, Canada. [44]Department of Medicine, University of Alberta, Edmonton, AB, Canada. [45]George and Fay Yee Centre for Healthcare Innovation, University of Manitoba, Winnipeg, MB, Canada. [46]Department of Pediatrics, The Hospital for Sick Children, University of Toronto, Toronto, Canada. [47]Department of Medicine, University of Toronto, Toronto, ON, Canada. [48]Kidney Transplant Program, Ajmera Transplant Centre, University Health Network, Toronto, ON, Canada. [49]Department of Medicine, Western University, London, ON, Canada. [50]Héma-Québec, Quebec, QC, Canada. [51]Division of Nephrology, University of Montreal, Montreal, QC, Canada. [52]Department of Pediatrics, University of British Columbia, Vancouver, BC, Canada. [53]Children's Hospital Oakland Research Institute, Oakland, CA, USA. [54]Department of Pathology and Laboratory Medicine, Emory University, Atlanta, GA, USA. [55]Department of Pathology and Laboratory Medicine, University of North Carolina, Chapel Hill, NC, USA. [56]Department of Medicine, Baylor Scott and White Health Systems, Temple, TX, USA. [57]Office of Clinical Pharmacology, US Food and Drug Administration, Silver Spring, MD, USA. [58]Institute of Systems, Molecular and Integrative Biology, University of Liverpool, Liverpool, UK. [59]Transplantation Immunology, Maastricht University Medical Center, Maastricht, The Netherlands. [60]Department of Clinical Pharmacy and Toxicology, Leiden University Medical Center, Leiden, The Netherlands. [61]Transplantation Immunology, University Hospital Heidelberg, Heidelberg, Germany. [62]Department of Clinical Pharmacology, University Medical Center Gottingen, Gottingen, Germany. [63]Pharmacology and Transplantation, INSERM, University of Limoges, Limoges, France. [64]School of Pharmacy, University of Otago, Dunedin, New Zealand. [65]Center for Health Technology Assessment, Semmelweis University, Budapest, Hungary.

