## [Peer Review File · Communications Biology]

Reviewers' comments:

Reviewer #1 (Remarks to the Author):

MINOR COMMENTS

Introduction section should be shortened, especially well-known information about HLA organisation.

Reviewer #2 (Remarks to the Author):

The manuscript describes the "reduction" of high resolution typing to eplets in order to simplify antibody epitope matching in organ transplantation. The approach and the manuscript is of interest, but falls short at many aspects.

Specific comments and recommendations

- 1) The authors report the simulation including eplets. Although mentioned in the materials and methods, no data were shown to validate their model with respect to the existing transplant allocation program. Benchmarking their model is crucial for the interpretation of the eplet simulation interpretations.
- 2) The authors typed their population of 2000 and executed simulations on allocation. As the sequences of many HLA alleles in known and many haplotype data are available, extensive simulations could be run on these data to further support the results from typing. This approach would allow generalization of the current findings and may suggest an applicability for allocation programs other than the Canadian one.
- 3) The authors suggest in their title that this NGS approach provides strategies for prospective HLA epitope matching. As the authors mention in their intro and discussion, epitopes exist for B cells and T cells. This manuscript only addresses the B cell epitopes. The title must be adapted.

Other suggestions and comments:

- 1) HLA typing has been described based upon a certain kit. It is unclear whether all loci were typed for all exons or whether this was done at the high resolution level. It is possibly reported by the vendor, but as this info is crucial for the interpretation, this should be mentioned in the text.

Reviewer #3 (Remarks to the Author):

The manuscript of Tran et al. presents interesting data on HLA eplets in a large Canadian transplant cohort. They provide a detailed analysis of eplets, e.g. how it is possible to partially reduce HLA complexity when converting alleles to eplets and frequencies distributions of alleles and eplets in recipients and donors. They also show as a concept of proof that prospective eplet matching is a feasible approach to minimize immunological risks prior to solid organ transplantation. The simulations provide useful information on the relative size of the wait lists that would optimize prospective eplet matching in national or regional transplant programs.

This manuscript presents convincing data with appropriate statistical analyses and will influence thinking and future practices in the fields of solid organ transplantation, histocompatibility and immunogenetics.

Hereafter is a list of points that remain to be addressed by the authors.

- The results on visualization of eplets on HLA proteins are interesting but not well integrated in the manuscript. It is not clear what the added value to the main conclusions is.
- It is also not well explained what is the added value of cluster analysis.

- I would recommend caution using the terminology "population", especially since the transplant cohort is described as ethnically very heterogeneous. Please amend to "transplant population" or "study cohort".
- Page 10, line 190: is there an inversion between genotypes and epitopes in the sentence? Based on Figure 6, I assume that there is a decreased number of epitopes compared to genotypes at HLA-DQA1, DQB1 and DPA1. Please clarify.
- Page 10, lines 203-205: DRB1*04 alleles are mentioned twice in the sentence, this is not clear. Also which are the alleles that have results consistent with linkage disequilibrium? Please amend the text.
- Page 14: The discussion briefly describes the mechanisms of allorecognition by T and B lymphocytes. Direct T-cell recognition is also an important component of allograft rejection (usually well controlled by immunosuppression). Compared to indirect T-cell epitopes defined by PIRCHE and B-cell epitopes predicted by HLAMatchmaker, direct T-cell recognition is less predictable and involves a large polyclonal response mediated by cross-reactive mechanisms. Nevertheless, the authors could add one sentence to the discussion acknowledging this pathway of allorecognition.
- Page 18, lines 365-368: insuring equal opportunity to recipient candidates during allocation is a crucial ethical parameter. The authors should develop the discussion on how this could be implemented in the context of prospective epitope matching since some recipients won't have matched donors despite the improvements compared to allele compatibility?
- Figures:
 - Figure 2 is not readable, we can only assume that the description in the manuscript is what we see, please improve or find another synthetic way to report these interesting results.
 - Figure 4: would it make sense to separate alleles per locus rather than regrouping class I and class II, respectively? This could help the reader better distinguish the profiles at each HLA locus. This is especially true for class II since eplets are not shared across loci (except for HLA-DR).
 - Figure 8: error bars are not visible (maybe because SD values are very small?).
 - Figure 9: a better choice of colors could help the reader distinguish between the loci.
 - Supplementary Figure 3: names of eplets are blurry, please improve resolution.

Responses to Reviewer Comments for Manuscript ID: COMMSBIO-20-2373A

Reviewer #1 (Remarks to the Author):

Comment 1: Introduction section should be shortened, especially well-known information about HLA organisation.

Response: We have shortened the Introduction by removing the detailed description of the HLA genes. *Introduction (page 3, lines 48 – 54).*

“Human Leucocyte Antigen (HLA) genes are the most polymorphic in the human genome with over 25,000 alleles now identified⁶. These genes code for highly immunogenic HLA protein isoforms expressed on nucleated cells and are known to be the principal transplantation antigens and primary targets of graft rejection⁷. Compatibility for HLA genes between donor organs and graft recipients ensures excellent outcome in live donor transplantation, and improves graft survival in deceased donor transplantation^{8,9}, but is difficult to achieve due to the heterogeneity of this gene complex.

Reviewer #2 (Remarks to the Author):

The manuscript describes the "reduction" of high-resolution typing to eplets in order to simplify antibody epitope matching in organ transplantation. The approach and the manuscript is of interest, but falls short at many aspects.

Specific comments and recommendations

Comment 1: The authors report the simulation including eplets. Although mentioned in the materials and methods, no data were shown to validate their model with respect to the existing transplant allocation program. Benchmarking their model is crucial for the interpretation of the eplet simulation interpretations.

Comment 2: The authors typed their population of 2000 and executed simulations on allocation. As the sequences of many HLA alleles in known and many haplotype data are available, extensive simulations could be run on these data to further support the results from typing. This approach would allow generalization of the current findings and may suggest an applicability for allocation programs other than the Canadian one.

Response (to both comments): Thank you for this recommendation, we concur fully that detailed and validated modelling of epitope-based allocation is vitally important.

However, the objectives of this report were to provide the first documentation of population frequencies of the defined sequence-derived antibody-determined HLA epitopes within a heterogeneous population of both donors and recipients, to show that these were closely comparable hence potentially minimizing matching imbalance, and based on these data to provide an initial base case model demonstrating the possibilities of achieving (or otherwise) eplet-matching at single or multiple gene loci within a national context, all of which data were previously unknown.

This first base-case model therefore included all patients actively awaiting transplantation in the province of B.C. (patients on hold status were eliminated; highly sensitized patients (PRA>95%) are listed on a separate national waiting list; and only a small number of patients are listed as urgent for medical reasons (normally < 5%). Deceased donor organs were allocated to these potential recipients according to ABO identity (the normal practice for routine transplantation), and recipients matched within the model were removed from the pool. No attempt was made at this stage to incorporate anti-HLA DSA, age matching, recipient or other factors.

With this knowledge in hand, we are now extending these studies in a larger sample of almost 2000 deceased donor / recipient transplanted pairs (2008-2020) allocated into test and re-test cohorts on whom we have completed full NGS typing and on whom we have full clinical, demographic and immunological data. This will enable the development and testing of a detailed model that will incorporate all cardinal donor and recipient factors employed in clinical organ allocation. The model will then be validated using a second comparable data set from another major provincial program (Quebec) to confirm precision and generalizability within Canada. We anticipate that this second article will be submitted by the end of the first quarter of 2021.

Comment 3: The authors suggest in their title that this NGS approach provides strategies for prospective HLA epitope matching. As the authors mention in their intro and discussion, epitopes exist for B cells and T cells. This manuscript only addresses the B cell epitopes. The title must be adapted.

Response: Thank you, we concur. The title has been changed to: “*NEXT-GENERATION SEQUENCING DEFINES DONOR AND RECIPIENT HLA B-CELL EPITOPE FREQUENCIES FOR PROSPECTIVE MATCHING IN TRANSPLANTATION*”.

Comment 4: HLA typing has been described based upon a certain kit. It is unclear whether all loci were typed for all exons or whether this was done at the high resolution level. It is possibly reported by the vendor, but as this info is crucial for the interpretation, this should be mentioned in the text.

Response: We have added the precise details regarding the genes and regions that were sequenced to the Materials and Methods.

Materials and Methods (page 34, lines 583 – 559).

“HLA gene sequencing, eplet analysis and carrier frequencies

DNA was extracted from whole blood using the EZ1 DNA Blood 350 µl Kit or QIAasymphony DSP DNA Mini Kit (192) (Qiagen, Hilden, Germany). Whole gene characterization (5'UTR to 3'UTR) for HLA-A, B, C, DQA1, DPA1, and DQB1 and key region characterization (exon 2 to 3'UTR) of HLA-DRB1, DRB3, DRB4, DRB5, and DPB1 was performed using the Holotype HLA kit version 2 (Omixon, Budapest, Hungary). Libraries were sequenced using MiSeq Sequencer (Illumina, California, USA) and sequence data analyzed using the HLA Twin version 2.5 (Omixon, Budapest, Hungary).”

Reviewer #3 (Remarks to the Author):

The manuscript of Tran et al. presents interesting data on HLA eplets in a large Canadian transplant cohort. They provide a detailed analysis of eplets, e.g. how it is possible to partially reduce HLA complexity when converting alleles to eplets and frequencies distributions of alleles and eplets in recipients and donors. They also show as a concept of proof that prospective eplet matching is a feasible approach to minimize immunological risks prior to solid organ transplantation. The simulations provide useful information on the relative size of the wait lists that would optimize prospective eplet matching in national or regional transplant programs.

This manuscript presents convincing data with appropriate statistical analyses and will influence thinking and future practices in the fields of solid organ transplantation, histocompatibility and immunogenetics.

Hereafter is a list of points that remain to be addressed by the authors.

Comment 1: The results on visualization of eplets on HLA proteins are interesting but not well integrated in the manuscript. It is not clear what the added value to the main conclusions is.

Response: The visualization of eplets on HLA proteins was included for readers who may be non-experts in this field. To improve the value of this visualization in the manuscript, we have adapted the figure and text to include a class I protein (A*02:01) and a class II protein (DQ2.3).

Results (page 5, lines 92 – 102).

“Visualization of eplets on HLA proteins

Eplets were mapped onto 3D HLA protein structures for class I (Figure 1a, 1b) and class II (Figure 1)13–15. In Figure 1a/b, HLA-A*02:01 is depicted from top-down and side views, respectively. Eplets existed in all extracellular regions, predominantly in the alpha helices around the peptide binding groove encoded by the key exons 2 and 3, but also outside this region in alpha 3 encoded by non-key exon 4. Figure 1c/d represents HLA-DQ2.3 protein, encoded by HLA-DQA*03:01 and HLA-DQB1*02:01. Eplets were found in the alpha helices of both alpha and beta chains, and within the beta sheet of the beta chain. Visualization of the side of the protein show eplets in beta 2 (encoded by non-key exon 3) of the beta chain. Thus, the majority of eplets are found in the peptide-binding region but eplets exist outside this area, showing that antibody binding can occur across the entire extracellular portion of the HLA protein.”

Figures (page 26, 502 – 511).

“Figure 1. 3D visualizations of class I and class II HLA proteins and their eplets. (a) Top-down view of the peptide-binding groove of HLA-A*02:01 is depicted (without β 2 microglobulin) in purple, a processed peptide in black, and eplets highlighted in yellow. (b) Side-view of HLA-A*02:01, showing alpha 3 and an eplet outside of the peptide-binding groove. (c) Top-down view of HLA-DQ is depicted with the DQA1*03:01 chain in fuchsia, DQB1*02:01 in aqua, a processed peptide in black, and eplets highlighted in yellow. (d) Side-view of HLA-DQ2.3 showing alpha 2 of the alpha and beta chains with highlighted eplets. Molecular graphics and analyses performed with UCSF Chimera, developed by the Resource for Biocomputing, Visualization, and Informatics at the University of California, San Francisco, with support from NIH P41-GM103311¹³ using the Protein Data Bank ID: 4d8p (HLA-A*02:01)¹⁴ and 4u6x (HLA-DQ2.3)¹⁵.”

Comment 2: It is also not well explained what is the added value of cluster analysis.

Response: Thank you for your comment. On reflection, we agree that the cluster analysis does not add significant value to the results and have removed this section and Supplemental Figure 3.

Comment 3: I would recommend caution using the terminology “population”, especially since the transplant cohort is described as ethnically very heterogeneous. Please amend to “transplant population” or “study cohort”.

Response: We agree and have modified all text that refers to the study group as “population” to either “transplant population” or “study population” (29, 73, 87, 205, 269, and 296).

Comment 4: Page 10, line 190: is there an inversion between genotypes and epitopes in the sentence? Based on Figure 6, I assume that there is a decreased number of epitopes compared to genotypes at HLA-DQA1, DQB1 and DPA1. Please clarify.

Response: Thank you for your observation. There was indeed an inversion between genotypes and epitopes and have corrected this in the text.

Results (page 10, lines 194 - 196).

“For DPA1, DQA1, and DQB1, 100% of epitopes occurred in patients and donors, with a significant decrease in the number of epitopes compared to genotypes.”

Comment 5: Page 10, lines 203-205: DRB1*04 alleles are mentioned twice in the sentence, this is not clear. Also, which are the alleles that have results consistent with linkage disequilibrium? Please amend the text.

Response: As recommended, we have removed the Cluster Analysis section and thus this sentence is no longer included in the text.

Comment 6: Page 14: The discussion briefly describes the mechanisms of allorecognition by T and B lymphocytes. Direct T-cell recognition is also an important component of allograft rejection (usually well controlled by immunosuppression). Compared to indirect T-cell epitopes defined by PIRCHE and B-cell epitopes predicted by HLAMatchmaker, direct T-cell recognition is less predictable and involves a large polyclonal response mediated by cross-reactive mechanisms. Nevertheless, the authors could add one sentence to the discussion acknowledging this pathway of allorecognition.

Response: Thank you. We have amended the first and second sentences of this section to include a reference to direct allorecognition, as well as both indirect and semi-direct (Alegre et al, 2016), and to emphasize the use of B-cell epitopes as recommended by Reviewer #2.

Discussion (page 13, lines 247 - 253).

“Complementary genomic and proteomic methods have clarified the structural biology of HLA antigens, enabling more precise understanding of the complex direct, indirect and semi-direct mechanisms of allorecognition by recipient lymphocytes¹⁶⁻¹⁹. Two cardinal groups of epitopes are now recognized, those involved in indirect recognition of the donor HLA antigen array by recipient T cell which are predicted through the PIRCHE algorithm²⁰, and those which are antibody-accessible and are involved in the humoral response (defined here as B-cell epitopes), predicted through the HLAMatchmaker algorithm¹⁰.”

Comment 7: Page 18, lines 365-368: ensuring equal opportunity to recipient candidates during allocation is a crucial ethical parameter. The authors should develop the discussion on how this could be implemented in the context of prospective epitope matching since some recipients won't have matched donors despite the improvements compared to allele compatibility?

Response: We have developed this section as recommended by including potential accommodations for recipients who will not have well-matched donors.

Discussion (page 17, lines 353 – 355).

“Potential accommodations include giving priority to these waitlisted patients (i.e. much like the Highly Sensitized Patients (HSP) Program), extending their minimum mismatch criteria, and setting a maximum time on the list until transplantation given all other criteria are met.”

Comment 8: Figure 2 is not readable, we can only assume that the description in the manuscript is what we see, please improve or find another synthetic way to report these interesting results.

Response: We have modified the figure to make it more readable by converting the allele names into dots, changing the colours, and rearranged the images. The figure is also provided in PDF form to allow for zooming in without loss of resolution. We have also referenced an animated version of the figure that readers can view for better visualization of the allele-eplet connections.

Results (page 6, line 113 – 114).

“An interactive version of the networks can be accessed at www.gctransplant.com.”

Figures (page 27, lines 512 – 516).

“Figure 2. Bipartite networks⁴⁴ depicting the associations between the complete library of HLA alleles and eplets present in HLAclassmaker v02 for HLA class I and v02.2 for HLA class II genes. Each node represents an allele and the lines connect them to eplets, showing that that allele expresses that eplet. Alleles are colour-coded by gene. Interactive visualizations of the networks can be accessed at www.genomecanadatx.com.”

Comment 9: Figure 4: would it make sense to separate alleles per locus rather than regrouping class I and class II, respectively? This could help the reader better distinguish the profiles at each HLA locus. This is especially true for class II since eplets are not shared across loci (except for HLA-DR).

Response: We chose to group alleles and eplets by class for better visualization, and still feel that this is the simplest and clearest presentation of the data. However, we have reformatted the figure as recommended and included it as Supplemental Figures 2 and 3 (available in PDF format for better visualization). If the reviewer prefers, we can exchange Figure 3 with these two figures.

Supplemental Figures.

“Supplemental Figure 3. The relative frequencies of HLA alleles in the study population categorized by class I genes and individual class II genes. Relative frequencies were calculated as the proportion of subjects expressing a particular allele. (A) Class I alleles, (B) DRB1, DRB3, DRB4, and DRB5 alleles, (C) DQA1 alleles, (D) DQB1 alleles, (E) DPA1 alleles, and (F) DPB1 alleles. KP = Kidney Patients, KD = Kidney Donors.”

“Supplemental Figure 4. The relative frequencies of HLA eplets in the study population categorized by class I genes and individual class II genes. Relative frequencies were calculated as the proportion of subjects expressing a particular eplet. (A) Class I eplets, (B) DRB1, DRB3, DRB4, and DRB5 eplets, (C) DQA1 eplets, (D) DQB1 eplets, (E) DPA1 eplets, and (F) DPB1 eplets. KP = Kidney Patients, KD = Kidney Donors.”

Comment 10: Figure 8: error bars are not visible (maybe because SD values are very small?).

Response: Yes, the error bars are small because the standard deviations of the simulation were very small. Thus, we therefore have not made changes to this figure.

Comment 11: Figure 9: a better choice of colours could help the reader distinguish between the loci.

Response: We agree and have modified the colours for better visualization.

Figures (page 33, lines 567 - 576).

“Figure 9. The effect of patient waitlist size on mismatch score in deliberative eplet matching simulations. Results were based on the averaged set of 10 deliberative eplet-matching simulations at various gene loci. Provincial active waitlist and deceased donor numbers were used according to Canadian Organ Replacement Register³. X axis shows the number of patients on provincial or national patient waiting lists (SK = Saskatchewan with 88 patients, Atlantic = Atlantic provinces (New Brunswick, Nova Scotia, and Prince Edward Island) with 141 patients, MB = Manitoba with 183 patients, BC = British Columbia with 207 patients, AB = Alberta with 250 patients, AveProv = Average across all provinces with 290 patients, QC = Quebec with 373 patients, ON = Ontario with 790 patients, and Canada with 2,032 patients). Y axis shows the averaged cumulative probability of achieving a total mismatch score of 0 (a) or 10 or lower (b) at the end of the simulations.”

Comment 12: Supplementary Figure 3: names of eplets are blurry, please improve resolution.

Response: As we have removed the Cluster Analysis section, we have also removed Supplementary Figure 3.

REVIEWERS' COMMENTS:

Reviewer #3 (Remarks to the Author):

The authors answered satisfactorily to my comments. I have no more comments.

**Responses to Reviewer Comments for Manuscript ID:
COMMSBIO-20-2373A**

Reviewer #1 (Remarks to the Author):

There were no comments from Reviewer #1 from the email sent by Dr. Grinham and Dr. Lui on February 8th, 2021.

Reviewer #2 (Remarks to the Author):

There were no comments from Reviewer #2 from the email sent by Dr. Grinham and Dr. Lui on February 8th, 2021.

Reviewer #3 (Remarks to the Author):

Comment 1: The authors answered satisfactorily to my comments. I have no more comments.

Response: Thank you for your comment and thus have not made any further changes.